# Cbf11 and Mga2 function together to activate transcription of lipid metabolism genes and promote mitotic fidelity in fission yeast

Anna Marešová[1], Michaela Grulyová[1], Miluše Hradilová[2], Viacheslav Zemlianski[1], Jarmila Princová[1], Martin Převorovský[1]*

1 Department of Cell Biology, Faculty of Science, Charles University, Prague, Czechia, 2 Laboratory of Genomics and Bioinformatics, Institute of Molecular Genetics of the Czech Academy of Sciences, Prague, Czechia

* prevorov@natur.cuni.cz

**Data Availability Statement:** The sequencing data are available from the ArrayExpress database

## Abstract

Within a eukaryotic cell, both lipid homeostasis and faithful cell cycle progression are meticulously orchestrated. The fission yeast *Schizosaccharomyces pombe* provides a powerful platform to study the intricate regulatory mechanisms governing these fundamental processes. In *S. pombe*, the Cbf11 and Mga2 proteins are transcriptional activators of non-sterol lipid metabolism genes, with Cbf11 also known as a cell cycle regulator. Despite sharing a common set of target genes, little was known about their functional relationship. This study reveals that Cbf11 and Mga2 function together in the same regulatory pathway, critical for both lipid metabolism and mitotic fidelity. Deletion of either gene results in a similar array of defects, including slow growth, dysregulated lipid homeostasis, impaired cell cycle progression (cut phenotype), abnormal cell morphology, perturbed transcriptomic and proteomic profiles, and compromised response to the stressors camptothecin and thiabendazole. Remarkably, the double deletion mutant does not exhibit a more severe phenotype compared to the single mutants. In addition, ChIP-nexus analysis reveals that both Cbf11 and Mga2 bind to nearly identical positions within the promoter regions of target genes. Interestingly, Mga2 binding appears to be dependent on the presence of Cbf11 and Cbf11 likely acts as a tether to DNA, while Mga2 is needed to activate the target genes. In addition, the study explores the distribution of Cbf11 and Mga2 homologs across fungi. The presence of both Cbf11 and Mga2 homologs in Basidiomycota contrasts with Ascomycota, which mostly lack Cbf11 but retain Mga2. This suggests an evolutionary rewiring of the regulatory circuitry governing lipid metabolism and mitotic fidelity. In conclusion, this study offers compelling support for Cbf11 and Mga2 functioning jointly to regulate lipid metabolism and mitotic fidelity in fission yeast.

## Author summary

Lipid metabolism and nuclear division are both needed to ensure proper cell growth and proliferation. Their connection has recently become an actively investigated

(https://www.ebi.ac.uk/arrayexpress/) under the accession numbers E-MTAB-13303 (RNA-seq of Δmga2 and Δmga2 Δcbf11) and E-MTAB-13305 (RNA-seq of WT, Pcut6MUT, WT treated with cerulenin and control WT treated with DMSO). Previously published sequencing data [doi:10.1242/jcs.261568] were obtained from ArrayExpress with the following accession numbers: E-MTAB-13258 for all ChIP-nexus libraries, E-MTAB-13302 and E-MTAB-13305 for RNA-seq of WT and Δcbf11. The scripts used for sequencing data processing and analyses are available from https://github.com/mprevorovsky/RNA-seq_ammonium (WT, Δcbf11), https://github.com/mprevorovsky/RNA-seq_CSL-DBM_cerulenin (WT, cerulenin treatment), and https://github.com/mprevorovsky/RNA-seq_mga2 (Δmga2, Δmga2 Δcbf11). Source data for Figures, including the numbers of independent experiments (n) in bar plots (Figs 1B, 2A, 3B, 3C, S1, S4, and S5) are available as S2 Data.

**Funding:** This work was supported by the Grant Agency of Charles University (GA UK grant no. 248120 to A.M.). M.H. was supported by the European Union - Next Generation EU (programme EXCELES project no. LX22NPO5102). The funders had no role in study design, data collection and analysis, decision to publish, or preparation of the manuscript.

**Competing interests:** The authors have declared that no competing interests exist.

phenomenon. Several studies in organisms ranging from yeasts to mammals showed that proper regulation of lipid metabolism is essential for accurate cell division. Cbf11 and Mga2 are two transcriptional regulators of genes involved in non-sterol lipid metabolism in the fission yeast *Schizosaccharomyces pombe*. While these proteins were previously thought to function independently of each other, our findings reveal that they cooperate closely to maintain lipid homeostasis and, in turn, promote the fidelity of nuclear division. Moreover, we demonstrate that Mga2 mirrors all analyzed functions of Cbf11, including control of cell cycle progression and maintenance of genome integrity. Our research provides novel insights into the intricate interactions between lipid metabolism and cell division.

## Introduction

Lipids are major constituents of cells, serving as energy reserves and membrane building blocks. In addition, they also have more specialized functions, e.g. in protein modification or as signaling molecules [1,2]. In yeasts, neutral lipids (triacylglycerols and steryl esters) are stored in cytoplasmic bodies called lipid droplets (LDs). LDs are generated from the endoplasmic reticulum and their hydrophobic lipid core is enclosed in a phospholipid monolayer with associated proteins [3,4]. When needed, neutral lipids can be mobilized from LDs to produce phospholipids or to enter energy-generating pathways [1,2].

The metabolism of lipids is regulated at multiple levels. In the fission yeast *Schizosaccharomyces pombe*, the phosphatidic acid phosphatase lipin (Ned1) serves as a switch between the production of neutral triglycerides and phospholipids, and thus plays a key role in regulating the production of new membranes [5]. This is especially critical in species undergoing closed mitosis, where rapid expansion of the nuclear envelope takes place during anaphase [6], and this expansion needs to be supported by sufficient phospholipid production [5,7,8]. When these requirements are not met, a catastrophic mitosis leading to the 'cut' phenotype can occur, with the undivided nucleus being untimely transected by the division septum [6,8–10].

At the transcriptional level, lipid homeostasis is maintained by two distinct regulatory branches. Sterol metabolism genes are activated by the membrane-tethered Sre1 transcription factor from the SREBP (sterol regulatory element binding protein) family. Sre1, the homolog of mammalian SREBP-2, is cleaved and activated upon sterol depletion or under hypoxia [11,12]. On the other hand, triglyceride metabolism genes are regulated by the Mga2 and Cbf11 transcription factors. Mga2, a functional analog of mammalian SREBP-1, is also expressed as a membrane-bound precursor in the endoplasmic reticulum and gets cleaved and activated upon fatty acid shortage or under hypoxia [13]. Interestingly, the budding yeast *Saccharomyces cerevisiae* contains two paralogs, Mga2p and Spt23p, which also regulate lipid metabolism genes [14]. Cbf11 belongs to the CSL (CBF1/Suppressor of Hairless/LAG-1) transcription factor family and it is a nuclear DNA-binding protein [15]. *S. pombe* cells lacking the *cbf11* gene display pleiotropic defects, including downregulated expression of fatty acid synthesis genes, aberrant LD content, and increased incidence of the 'cut' phenotype [15,16]. Notably, CSL family members are absent from the budding yeast [17]. Intriguingly, there is crosstalk between the sterol and triglyceride regulatory branches, as both Sre1 and Mga2 activation is regulated by the abundance of products of the other branch [18].

The Cbf11 and Mga2 regulators have been described independently, but have been shown to display some overlapping functions. Namely, they share a common set of lipid-related target genes, such as the *cut6* acetyl-CoA carboxylase and *ole1* acyl-CoA desaturase, and deletion

mutants in both genes show impaired growth and sensitivity to hypoxia [13,15,16]. Despite these similarities, little is known about whether and how these two proteins cooperate. The aim of this study is to explore the functional relationship between Cbf11 and Mga2 in the transcriptional regulation of lipid metabolism, and in promoting mitotic fidelity. Importantly, our results demonstrate that Cbf11 and Mga2 are not independent regulators, but rather function together to activate transcription of lipid metabolism genes and promote orderly mitotic progression.

## Results

### Cells lacking the *cbf11* and/or *mga2* gene display similar, non-additive growth and cellular defects

Deletion of the *cbf11* or *mga2* transcription factor gene leads to impaired growth [13,15]. Moreover, we have recently shown that both *Δcbf11* and *Δmga2* mutants exhibit increased expression of stress genes and are resistant to $H_2O_2$ [19]. To investigate any potential genetic interactions between *cbf11* and *mga2*, we first created *Δcbf11* and *Δmga2* single, and *Δmga2 Δcbf11* double deletion strains in an isogenic background. As expected, the new single deletion strains showed the previously reported growth defects. Interestingly, these growth defects were not additive as the *Δmga2 Δcbf11* double deletion strain did not show any further exacerbation of slow growth when cultured in the complex YES medium (S1 Fig). Previously, we showed that numerous defects of *Δcbf11* cells, including growth defects, are nutrient-dependent and can be suppressed by cultivation in the minimal EMM medium, in which ammonium chloride is the nitrogen source [16,20]. Therefore, we asked whether the growth of *Δmga2* cells also responds to nutrients. Indeed, the doubling time of both *Δmga2* and *Δmga2 Δcbf11* cultures was also reduced in EMM, in a manner similar to *Δcbf11* cells (S1 Fig).

Cbf11 and its DNA-binding activity are required for orderly cell cycle progression. Cells lacking the *cbf11* gene or cells with a point mutation that disrupts DNA binding of Cbf11 (*cbf11DBM*) exhibit various cellular and nuclear defects, such as the 'cut' phenotype, fragmented nuclear mass, aberrant septation and/or cell separation [15,21]. Therefore, we next determined whether any such defects were also present in cells lacking *mga2*. To this end, we analyzed WT, *Δcbf11*, *Δmga2*, and *Δmga2 Δcbf11* cells stained with DAPI by microscopy. Strikingly, the *Δmga2* mutant largely recapitulated the morphological and cell-cycle defects of the *Δcbf11* cells (Fig 1A and 1B). Interestingly, the incidence of the 'cut' phenotype in the double deletion mutant was not further increased compared to the single mutants (Fig 1B). In summary, we have shown that disruption of either of the two transcriptional activators of lipid metabolism genes leads to cell-cycle defects, and these are not additive in the *Δmga2 Δcbf11* double mutant.

### Cells lacking the *cbf11* or *mga2* gene display similar transcriptomic and proteomic signatures

To examine the possible joint functioning of Cbf11 and Mga2, we first determined the transcript levels of their selected target genes in *Δcbf11*, *Δmga2*, and *Δmga2 Δcbf11* cells by RT-qPCR. The panel of tested genes included *cut6* (acetyl-CoA carboxylase), *ole1* (acyl-CoA desaturase), *lcf1* and *lcf2* (long-chain fatty acid-CoA ligases), *fsh2* (serine hydrolase-like), *vht1* (biotin transporter), *bio2* (biotin synthase) and *fas2* (fatty acid synthase alpha subunit). We observed that all tested genes were significantly downregulated in both single mutants compared to WT (Fig 2A). Interestingly, the decrease in transcript levels in the double deletion mutant was not additive compared to the single mutants (Fig 2A). Next, we confirmed and extended these results by an RNA-seq analysis (Figs 2B and S2). We found a strong overlap

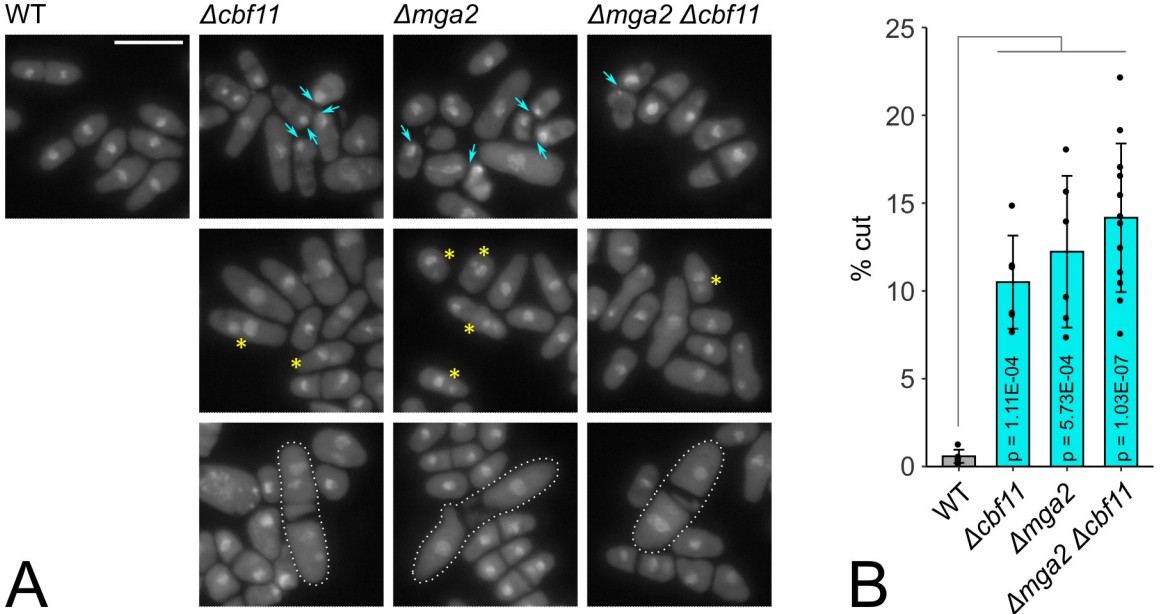

**Fig 1. Cells lacking Cbf11 and/or Mga2 display similar, non-additive cellular defects. (A)** Cells of the indicated genotypes were stained with DAPI to visualize nuclei. Cyan arrows point to the cells displaying the 'cut' phenotype and indicate positions where a nucleus is trapped in a septum and/or cell wall (top row); cells with fragmented and/or diffuse nuclei are marked with yellow asterisks (middle row); filamentous cells with septation defects are highlighted with dotted outlines (bottom row). Representative images from 6 independent experiments are shown. Scale bar represents 10 μm. **(B)** Quantification of 'cut' phenotype occurrence. Mean ± SD values, as well as individual data points for ≥6 independent experiments are shown. Significance was determined by a Welch Two Sample t-test.

between the sets of differentially expressed genes (DEGs) in the *Δcbf11* and *Δmga2* single mutants (Fig 2C). In addition, the double deletion mutant showed a transcriptomic profile almost identical to *Δmga2* (Fig 2C).

We have previously shown that deletion of the *cbf11* gene or disabling Cbf11 binding to DNA (Cbf11DBM) leads to upregulation of many non-coding RNAs ([21]; Fig 2E). We wanted to know whether this phenotype is specific to the Cbf11 regulator or whether it is more generally related to perturbed lipid metabolism. To this end we analyzed the non-coding transcriptomes of *Δcbf11*, *Δmga2*, and *Δmga2 Δcbf11* cells, and also included the *Pcut6MUT* lipid metabolism mutant, and WT cells treated with cerulenin. In the *Pcut6MUT* strain, the *cut6* promoter lacks the Cbf11 binding site and *cut6* expression drops to 50% [22], while cerulenin is a fatty acid synthase inhibitor. Strikingly, we found very similar deregulation of the non-coding transcriptome in all tested mutants and treatments (Figs 2E and S3). Thus, we conclude that unperturbed lipid metabolism is required for correct expression of the non-coding transcriptome.

Finally, since the absence of either the *cbf11* or *mga2* gene resulted in similar transcriptome changes, we decided to also compare the proteomes of the *Δcbf11* and *Δmga2* mutants using untargeted mass spectrometry. This analysis revealed a dramatic overlap of proteins that were expressed differentially in the respective single mutants compared to WT (Fig 2D). Taken together, these results strongly suggest that the functions of Cbf11 and Mga2 are very closely related.

## Cbf11 and Mga2 have overlapping roles in maintaining lipid homeostasis

Cbf11-deficient cells show an overall reduction in the abundance of lipid droplets (LDs) and pronounced cell-to-cell LD heterogeneity when grown in YES [16,21]. Both these phenotypes

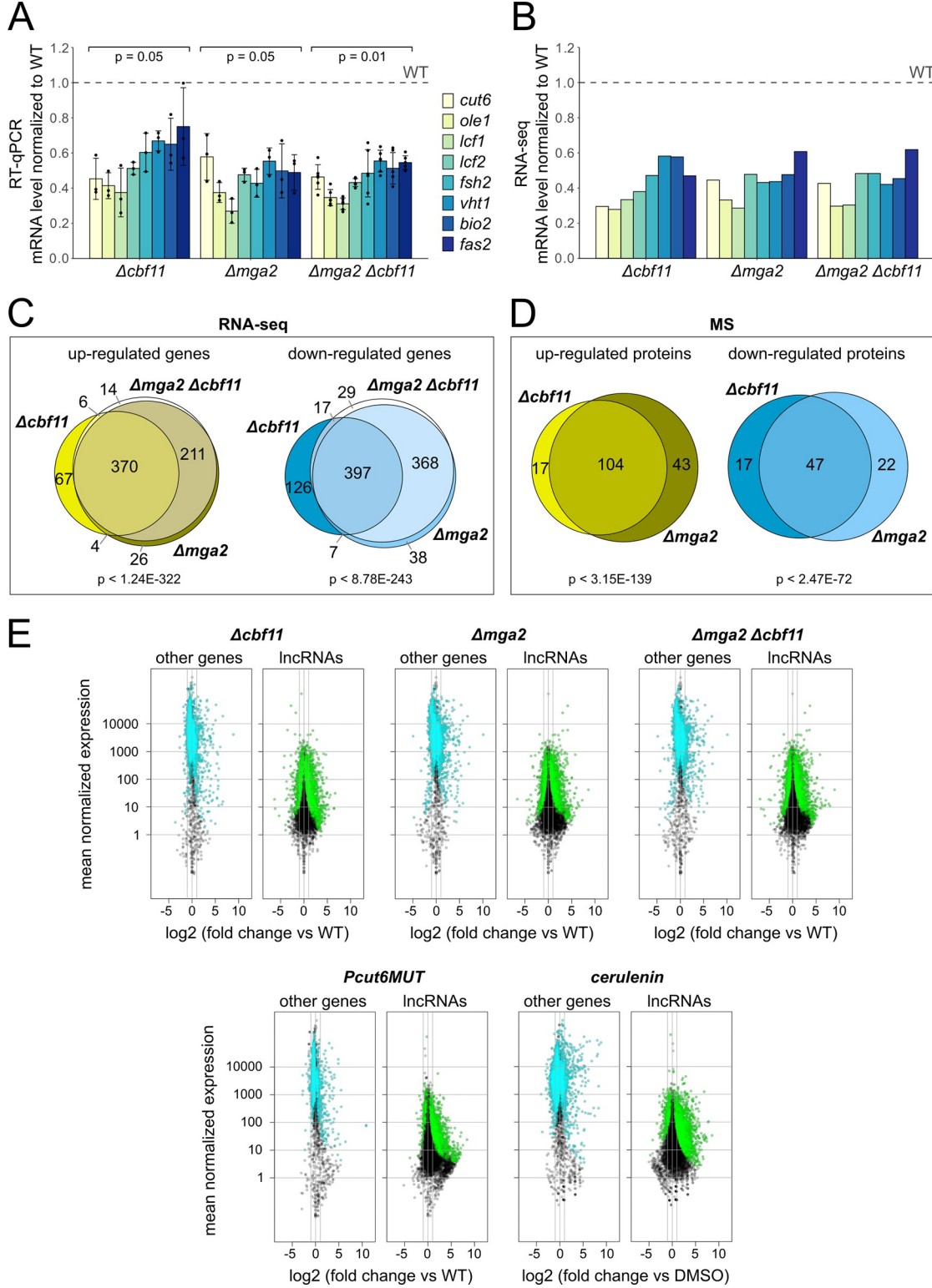

**Fig 2. The absence of Cbf11 and/or Mga2 results in similar transcriptome and proteome changes.** RT-qPCR (**A**) and RNA-seq (**B**) show that lipid metabolism genes are downregulated in *Δcbf11* and/or *Δmga2* mutants and this effect is not additive in the double deletion mutant. Means values ± SD, as well as individual data points for ≥3 independent experiments are shown in panel (**A**). Mean values of 3 independent RNA-seq experiments are shown in panel (**B**). Significance of changes was determined by a one-sided Mann-Whitney U test. Venn diagrams showing that the sets of genes (**C**) and proteins (**D**) expressed differentially in *Δcbf11*

and/or *Δmga2* cells significantly overlap (Fisher's exact test of RNA-seq and MS proteomics results, respectively). Only protein-coding genes and proteins showing at least 1.5-fold or 2-fold and statistically significant change in expression compared to WT are included in panel (C) and (D), respectively. **(E)** Both genetic and chemical perturbation of lipid metabolism result in deregulation of the non-coding transcriptome. Gene expression was determined by RNA-seq and compared to WT. Three independent experiments were performed. For each gene, the magnitude of change in expression was plotted against its mean expression value across all samples. Statistically significant differentially expressed genes are shown in cyan (other genes) or green (lncRNAs). The two gray vertical lines indicate the interval of 2-fold up- or downregulation.

are rescued by growing the mutant cells in EMM [20]. Therefore, we now also analyzed LD content in the *Δmga2* single, and *Δmga2 Δcbf11* double deletion mutants (Fig 3). Similar to the *Δcbf11* mutant, we observed a decrease in LD content (in both LD staining intensity and LD number per unit of cell volume; Fig 3B and 3C), and marked LD content heterogeneity in cultures lacking Mga2 grown in YES (Fig 3A). Notably, the defects were not further exacerbated in the double mutant. Furthermore, in all mutants the defects were largely suppressed when cells were grown in EMM. So, once again, the phenotypes of the *Δcbf11* and *Δmga2* mutants were very similar and were non-additive, suggesting joint functioning of Cbf11 and Mga2 in lipid homeostasis.

## Cbf11 and Mga2 are both required for genome integrity

We have recently described a requirement for Cbf11 in the maintenance of chromatin structure and genome integrity. Cells lacking intact Cbf11 are sensitive to the microtubule poison thiabendazole (TBZ), and to camptothecin (CPT), an inhibitor of topoisomerase II and a potent inducer of double-stranded DNA breaks [21,23]. We now tested whether Mga2 is also involved in these processes. Indeed, both *Δmga2* and *Δmga2 Δcbf11* mutants showed sensitivities to TBZ and CPT similar to the *Δcbf11* mutant (Figs 4 and S4). So far, our analyses revealed that the *Δmga2* mutant is very similar to *Δcbf11* cells, and that the phenotypes of the *Δmga2 Δcbf11* double mutant are non-additive. Therefore, our results strongly suggest joint functioning of Cbf11 and Mga2 in a range of cellular processes.

## Cbf11 and Mga2 share DNA binding sites

We have recently mapped Cbf11 binding sites using the genome-wide ChIP-nexus technique [21], which can determine the target protein footprint on DNA at a much higher resolution compared to traditional ChIP-seq [24]. Since Mga2 is also presumed to be a DNA-binding transcription factor, we set out to conduct a ChIP-nexus analysis for Mga2 as well. Cbf11 and Mga2 regulate a highly overlapping set of genes ([13,16]; Fig 2), so we were especially interested in determining how this joint regulation is carried out. Cbf11 is known to be able to bind DNA directly [25]. Therefore, we considered the following three scenarios (Fig 5A): a) Cbf11 and Mga2 bind to target promoter DNA individually and independently of each other, b) they bind DNA together as a complex, with each protein contacting the promoter via its own DNA-binding domain, c) they bind DNA together as a complex, with Cbf11 serving as the DNA-binding subunit. To avoid as many confounding variables as possible, we used isogenic scarless TAP knock-in strains constructed using CRISPR/Cas9. The C-terminally tagged Cbf11-TAP strain has been described previously [19,21]. Inactive, membrane-bound Mga2 undergoes proteolytic activation that liberates its N-terminal part, which can then regulate transcription in the nucleus [18]. Therefore, the TAP tag was fused to the N-terminus in the case of Mga2. Next, we also deleted the *cbf11* and *mga2* gene in the TAP-Mga2 and Cbf11-TAP background, respectively, to allow for assessing the interdependence of Cbf11 and Mga2 for binding to DNA. We confirmed that the introduction of the TAP tag had no adverse effect on the growth rate. The growth of the TAP-Mga2 *Δcbf11* strain was only slightly slower compared

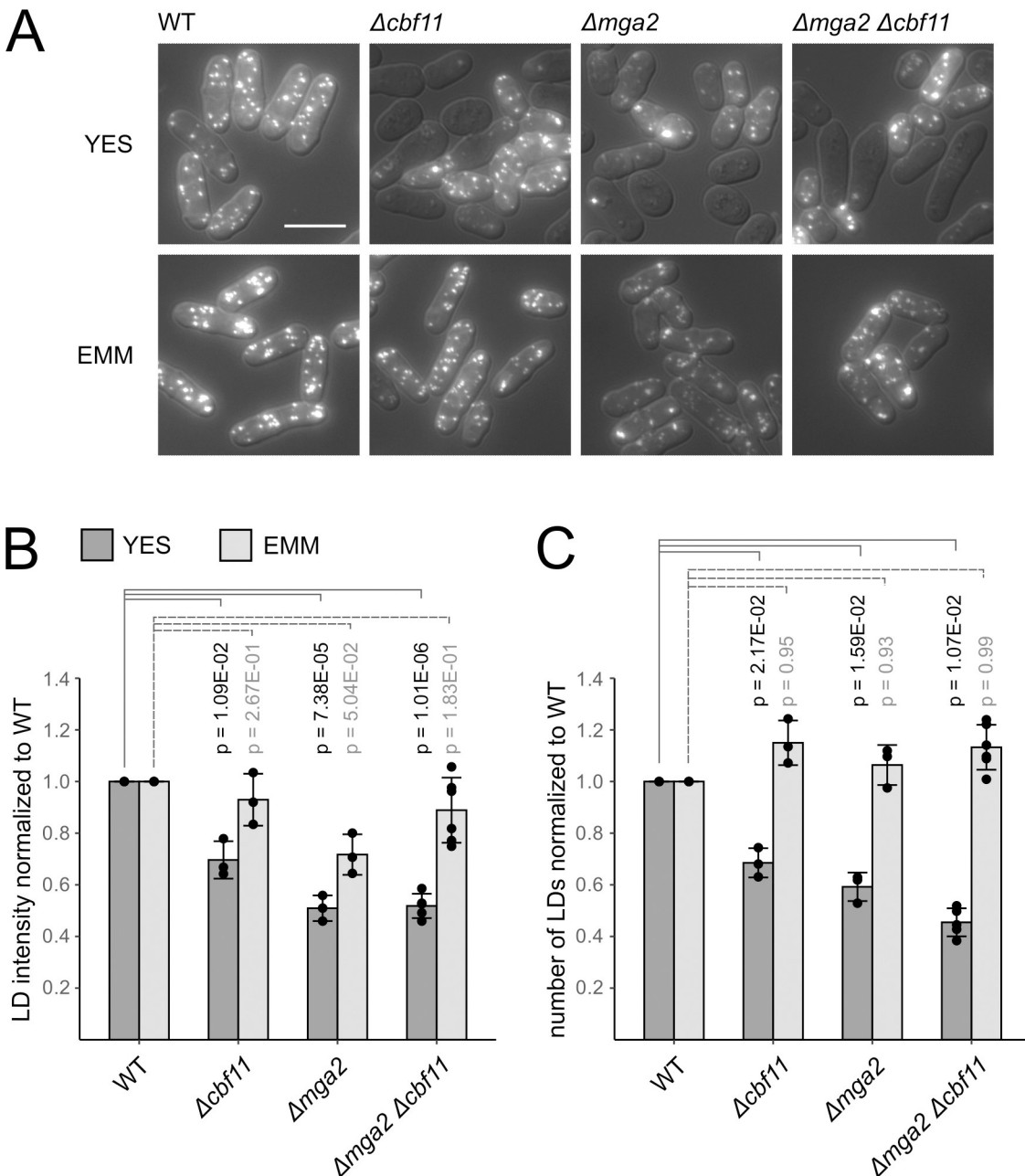

**Fig 3. Cells lacking Cbf11 and/or Mga2 display nutrient-responsive decrease in lipid droplet (LD) content. (A)** Cells of the indicated genotypes were stained with BODIPY 493/503 to visualize neutral lipids. Note the heterogeneity in LD content in *Δcbf11* and/or *Δmga2* mutants grown in the complex YES medium. The aberrant LD content is largely suppressed when cells are grown in the minimal EMM medium. Representative microscopic images from 3 independent experiments are shown. DIC overlay is shown to mark cell boundaries. Scale bar represents 10 μm. **(B, C)** Quantification of LD staining intensity and LD numbers per unit of cell volume. Mean ± SD values, as well as individual data points for ≥3 independent experiments are shown. Significance was determined by a Welch Two Sample t-test.

to untagged *Δcbf11*, while the growth of the Cbf11-TAP *Δmga2* strain was comparable to the growth of untagged *Δmga2* (S5 Fig).

Subsequent ChIP-nexus analysis revealed that the exact positions of the binding sites of Cbf11 and Mga2 in the promoters of most lipid metabolism genes overlap closely and show

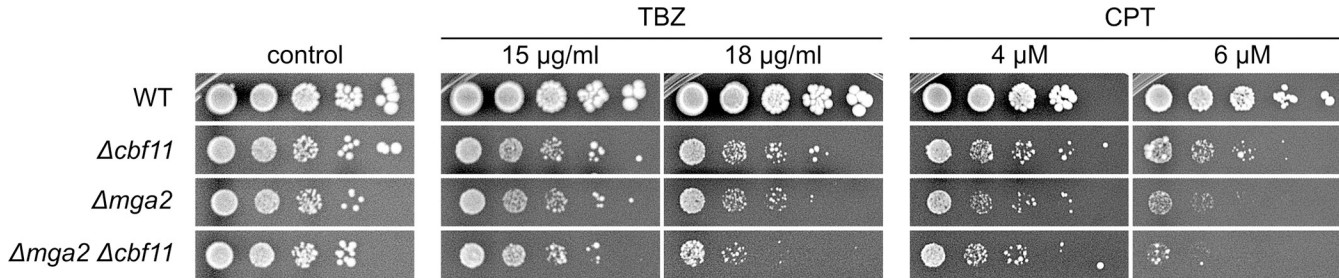

**Fig 4. Cells lacking Cbf11 and/or Mga2 are sensitive to thiabendazole (TBZ) and camptothecin (CPT).** Ten-fold serial dilutions of cultures of the indicated genotypes were plated on YES medium containing TBZ or CPT. Representative images from 2 independent experiments are shown.

very similar footprints (Figs 5B and S6). Moreover, the binding of Mga2 to DNA at most sites was dependent on Cbf11, as Mga2 in the *Δcbf11* background did not bind to the promoters of *cut6*, *ole1* (Fig 5B), *lcf2*, *ptl1*, *fsh2* and *vht1* (S6A Fig). Nevertheless, we also observed cases where the Cbf11 and Mga2 binding patterns were different (S6B Fig): a) Mga2 binding to the *lcf1* promoter was Cbf11-dependent, but then the absence of Mga2 resulted in markedly reduced binding and an altered footprint of Cbf11, b) both Cbf11 and Mga2 binding to the *fas1* promoter was fully dependent on the presence of the other counterpart, c) the *ptl2* promoter was bound by Cbf11 or Mga2 regardless of the presence of the other protein, even though the occupancy was reduced when the other protein was missing. We also found that the promoters of *bio2* and *fas2*, genes whose expression is Cbf11 and Mga2 dependent (Fig 2A and 2B), were not bound by Cbf11 or Mga2 proteins at all, suggesting indirect regulation (S6C Fig). Overall, although we observed several different binding arrangements, the majority of target promoters showed closely overlapping binding profiles of Cbf11 and Mga2, and this binding to DNA was dependent on Cbf11.

Finally, we compared the Cbf11 and Mga2 ChIP-nexus results with results of transcriptome analyses of *Δcbf11* and *Δmga2* deletion mutants. We found that binding of Cbf11 and Mga2 to the promoters of numerous genes of lipid metabolism (*cut6*, *ole1*, *lcf1*, *lcf2*, *fsh2*, *vht1*, *ptl2*, *fas1*; Figs 5B, S6A, and S6B) correlated with reduced transcript levels of these genes when either *cbf11* or *mga2* was deleted (Figs 2A, 2B and S2). Thus, while Cbf11 can bind the target promoters on its own, the presence of Mga2 is then required to bring about changes in target gene transcription. Taken together, these results are compatible with a scenario where Cbf11 and Mga2 form a complex that activates the expression of lipid metabolism genes. In this hypothetical complex, Cbf11 would serve as the DNA-binding subunit, while Mga2 would serve as the transcription activation subunit (see Fig 5A, bottom scenario). Nevertheless, we were unable to detect a physical interaction between Cbf11 and Mga2 (see Discussion).

## Mga2 has both CSL-dependent and independent functions in Fungi

All our results so far indicated that Cbf11 and Mga2 function together in *S. pombe*. Curiously, the budding yeast *Saccharomyces cerevisiae* is known to contain two Mga2 homologs (Mga2p and Spt23p; [14]), while no CSL family members are present [17]. Similar to *S. pombe*, the budding yeast Mga2p/Spt23p are also implicated in regulating lipid metabolism genes such as the fatty acid desaturase *OLE1* [14], suggesting that Mga2 homologs can also function in a CSL-independent manner. To gain more insight into the evolutionary rewiring of the Mga2-CSL circuitry, we examined the phylogenetic distribution of these two protein families within the Fungi kingdom. Using the protein sequences of *S. pombe* Cbf11 and Mga2 as queries, we found 109 taxons that contained both Cbf11 and Mga2 homologs, 386 taxons that contained

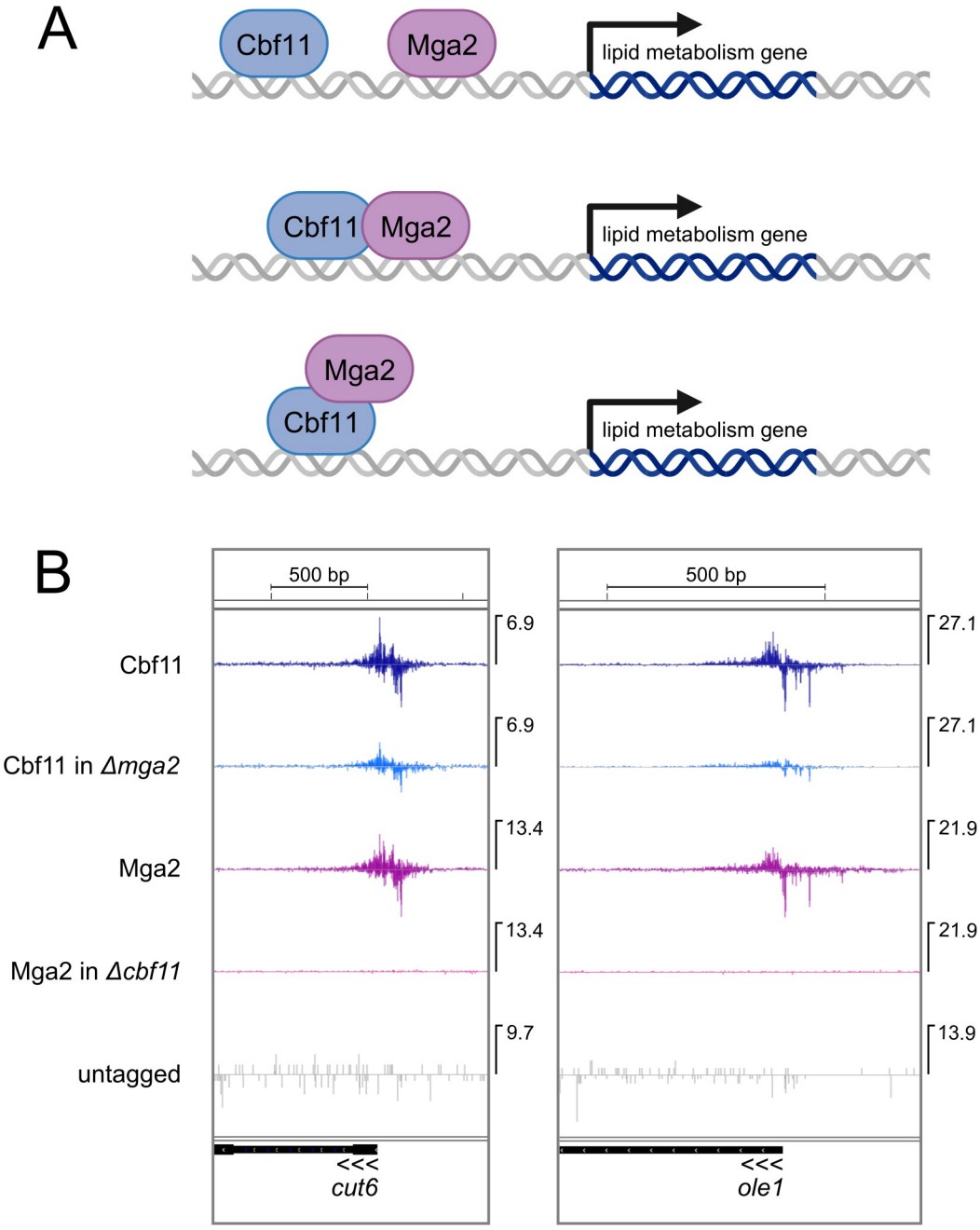

**Fig 5. Mga2 binds to DNA in a Cbf11-dependent manner. (A)** Scheme of three considered DNA binding regimes of Cbf11 and Mga2. See the main text for more detailed description. Image created with BioRender.com. **(B)** *In vivo* binding of Cbf11 and Mga2 TAP-tagged proteins to the promoters of selected lipid metabolism genes was analyzed by ChIP-nexus. Mean strand-specific coverage profile of 3 independent experiments for Cbf11 and Mga2 in the indicated genetic backgrounds, and a strand-specific coverage profile for untagged WT cells (negative control) are visualized in the Integrated Genome Viewer (IGV; Broad Institute). Gene orientation is indicated by three arrowheads; the maximum Y-axis value for each track is indicated on the right side of each panel.

only Mga2 homolog(s), and 1 taxon that contained only Cbf11 homolog(s) (Table E in S1 Data). A phylogenetic tree was then generated from all identified taxons (Fig 6) to assess the distribution of Cbf11 and Mga2 homologs.

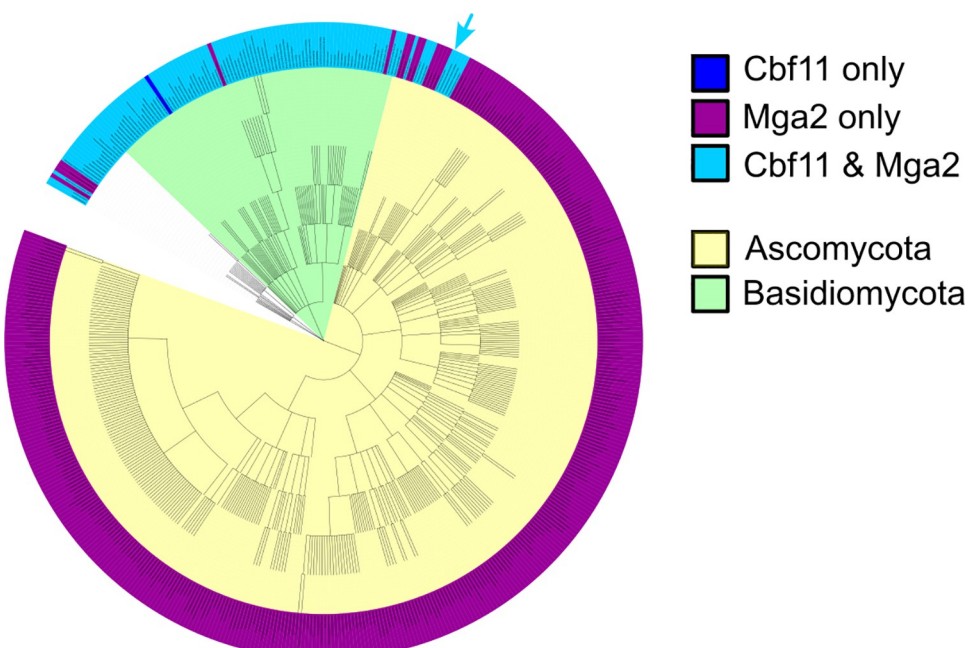

**Fig 6. Phylogenetic distribution of Cbf11 and Mga2 homologs in Fungi.** In the phylum Basidiomycota, members of the Cbf11 and Mga2 families are found together in almost all species. However, in the phylum Ascomycota, Cbf11 homologs are typically absent, except for the Taphrinomycotina group to which the fission yeasts belong (*S. pombe* position indicated with a light blue arrow). The taxonomy of some fungal species is uncertain (white part of the tree). Image created with iTOL v6.

We found that in the Basidiomycota phylum the CSL and Mga2 families co-occur in most taxons (Fig 6). The only exceptions were one species from the Ustilaginomycotina subdivision (*Ceraceosorus guamensis*) that apparently lacks Mga2 homologs, and two species from the Agaricomycotina subdivision (*Tremella mesenterica* and *Mycena indigotica*) that do not seem to contain any Cbf11 homologs. However, these rare exceptions may be false negatives and merely reflect potential errors in genome assembly of the respective species. By contrast, the CSL/Mga2 distribution was markedly different in the Ascomycota phylum. Here, Mga2 homologs were universally present, but Cbf11 homologs were typically absent (Fig 6). More specifically, CSL proteins were only found in several species belonging to the Taphrinomycotina subdivision. These included members of the class *Schizosaccharomycetes* (*Schizosaccharomyces pombe*, *S. japonicus*, *S. octosporus*, *S. cryophilus*, *S. osmophilus*), the class Pneumocystidomycetes (*Pneumocystis murina*, *P. jirovecii*, *P. carinii*), and the class Taphrinomycetes (*Protomyces lactucae-debilis* and *Saitoella complicata*). Our results thus suggest that in fungi Cbf11 homologs occur and likely also function (almost) exclusively together with Mga2 homologs, whereas Mga2 proteins can operate independently of the CSL protein family.

## Discussion

Mga2 and the CSL family transcription factor Cbf11 are currently the only known transcriptional regulators of fatty acid and triacylglycerol metabolism in the fission yeast *S. pombe* [13,22]. They were shown to regulate a common set of genes [13,16], but nothing was known about the potential functional interactions between Mga2 and Cbf11. To decipher the relationship between Cbf11 and Mga2, we now subjected strains lacking *cbf11* and/or *mga2* to extensive functional analyses. In all test cases, the behavior of *Δmga2* corresponded to the

phenotype of *Δcbf11*. Moreover, the double deletion mutant did not show an additive phenotype. We showed that the positions of the Cbf11 and Mga2 binding sites closely overlap in most promoters of their target lipid metabolism genes. Typically, Mga2 cannot bind to DNA in the absence of Cbf11. On the other hand, Cbf11 can bind to DNA on its own, but it cannot activate transcription without Mga2. Thus, our results demonstrate that Cbf11 and Mga2 most likely function in the same regulatory pathway. Finally, we have established Mga2 as a novel regulator of mitotic progression, as *Δmga2* cells are prone to the 'cut' catastrophic mitosis. It is tempting to speculate that Mga2 contributes to regulation of mitotic progression in the same way as Cbf11 does, since Mga2 mirrors all tested functions of Cbf11, as shown in this study. We have recently shown that proper cohesin dynamics and chromatin structure, i.e. factors that are altered in the *Δcbf11* mutant, contribute to mitotic fidelity [23].

The data we gathered strongly suggested a possibility that Cbf11 and Mga2 form a protein complex. Also, Mga2 was shown previously to co-purify with Cbf11-TAP in a one-step IgG-based pull-down followed by mass spectrometry [26]. However, in the current study we failed to detect a physical interaction between Cbf11 and Mga2, despite using a range of affinity purification approaches of both endogenous and heterologously expressed proteins. This may be a false negative result caused by the low abundance of both proteins (hundreds of molecules per cell; [27,28]) or their susceptibility to proteolysis [29]. Also, the hypothetical physical interaction between Cbf11 and Mga2 may be too transient to be detected by biochemical purifications. To further test the suspected Cbf11-Mga2 physical interaction, future studies could therefore employ alternative methods such as proximity-dependent biotinylation assay (TurboID; [30]), fluorescence resonance energy transfer (FRET; [31]), or the yeast two-hybrid system [32]. Alternatively, Cbf11 and Mga2 could cooperate via a different mechanism altogether, such as Cbf11 first binding to promoter DNA and affecting local chromatin structure to allow subsequent Mga2 recruitment and gene activation without the need for physical interaction between Cbf11 and Mga2.

The production, utilization and/or storage of various lipid types needs to adapt to factors such as nutrient availability, population density, osmotic pressure, ambient temperature, etc. [1,2]. It is conceivable that Cbf11 and Mga2 activities are coupled to these environmental cues to achieve proper lipid homeostasis. Indeed, Mga2 is required for growth under hypoxic conditions, where low oxygen availability limits the process of fatty acid desaturation, thereby affecting membrane rigidity and vesicle trafficking [18]. Interestingly, the proteolytic activation of the membrane-bound Mga2 precursor is triggered by fatty acid shortage, and Mga2 activity was also found to be cross-regulated by the Sre1/SREBP-dependent sterol synthesis pathway [18]. Moreover, cells lacking Cbf11 are sensitive to low temperatures [15], a condition where proper adjustments of membrane composition are critical for survival. Finally, we have shown previously [20,33] and also in the present study that Cbf11 and Mga2 activities are affected by nitrogen availability. A well-utilizable nitrogen source activates the Tor2 kinase, a major growth regulator of *S. pombe* [34]. Intriguingly, both Cbf11 and Mga2 are phosphorylated at multiple sites [27,35,36], and the phosphorylation at S505 and T1091 of Mga2 is actually TOR-dependent [37]. On top of that, Cbf11 and Mga2 are sumoylated at K89 and K615, respectively [38]; Mga2 is glycosylated at N17 and N998 [39], and ubiquitinated at multiple positions [40,41]. While the biological significance of these numerous posttranslational modifications remains unknown, they represent plausible means of regulating Cbf11 and Mga2 protein abundance and activity. Indeed, ubiquitination at specific lysine residues is important for the proteasome- and Cdc48p ATPase-dependent activation of the Mga2p120 precursor in *Saccharomyces cerevisiae* [42].

Our phylogenetic analyses showed that Mga2 homologs operate in a CSL-independent manner in the vast majority of ascomycetes (Fig 6). A question thus arises how these Mga2

homologs bind to DNA—do they utilize a CSL-unrelated DNA binding or recruiting partner, or can they bind to DNA directly? Earlier studies in *S. cerevisiae* identified the LORE (low oxygen response element) DNA motif in the promoter of the *OLE1* fatty acid desaturase gene [43], a regulatory target of Mga2p/Spt23p [44]. Notably, the core LORE sequence ACTCAACAA is not reminiscent of the Cbf11 response element CGTG(G/A)GAA. Interestingly, *S. cerevisiae* Mga2p/Spt23p are not thought to be capable of direct DNA binding [45], but a LORE-binding protein complex has been identified which does contain Mga2p [46]. This suggests that ascomycetal Mga2 homologs (with the exception of Taphrinomycotina) serve as transcriptional co-activators that require a yet unidentified DNA-binding partner(s) for accessing target promoters.

It should also be noted that *S. pombe* contains a second CSL protein, Cbf12 [15], which also functions as a DNA-binding activator of transcription [25,29,47]. However, Cbf12 has a specialized function in cell adhesion [15,47], and its deletion does not cause any adverse phenotype under standard growth conditions [15], unlike in the case of Cbf11 or Mga2. There are currently no indications that Cbf12 cooperates with Mga2, but the exact mechanism of Cbf12-mediated transcription activation, including whether Cbf12 requires a coactivator partner, remains to be established.

Finally, it is intriguing that while the ablation of Cbf11 and/or Mga2 results in a plethora of cellular defects, the presence of both Cbf11 and Mga2 is not strictly essential for viability, even though they regulate essential genes (such as *cut6*) and processes. This can be explained by the fact that in the absence of both Cbf11 and Mga2 the expression of their target genes only drops by ~50% (Fig 2). It is therefore possible, or perhaps even likely, that *S. pombe* possesses additional activator(s) of non-sterol lipid metabolism genes, which act independently and in parallel to Cbf11 and Mga2. These remain to be identified by future studies.

## Materials and methods

### Plasmid construction

The Cas9/sgRNA_*mga2* plasmid (pMP167) was prepared as follows. sgRNA targeted to the *mga2* ORF was inserted into the pLSB plasmid (alternative name pMP160) carrying an empty sgRNA cassette (with a GFP placeholder) and a Cas9 endonuclease gene that is codon-optimized for expression in *S. pombe* [48]. Briefly, a suitable sgRNA targeting the 5' end of the *mga2* ORF was manually designed, 24-nt forward and reverse sgRNA oligonucleotides (AJ89 and AJ90; 20-nt sgRNA with 4-nt *BsaI* overhangs) were annealed, phosphorylated and inserted into the *BsaI*-digested pLSB plasmid.

To create the pMP168 plasmid, the *TAP-mga2*\* fragment for HR (see S1 Text) was cloned into the pBluescript SK(+) plasmid. Briefly, HR template was divided into 3 parts and each part was amplified by NEB Q5 polymerase using AJ51 & AJ52 oligonucleotides for part 1, AJ53 & AJ54 for part 2, and AJ55 & AJ56 for part 3. Genomic DNA from WT cells was used as a template for PCR amplification of HR part 1 and 3. To amplify HR part 2, pMaP27 plasmid containing a TAP-tag sequence was used as template. All three PCR-amplified and purified parts were inserted into the pBluescript SK(+) vector to create the final pMP168 plasmid. Note that the HR template contains a point mutation (marked as \*) that disrupts the PAM sequence in the *mga2* ORF to avoid re-cleavage by Cas9.

Both final plasmids pMP167 and 168 were verified by restriction cleavage and sequencing.

### Cultivations, media and strains

*Schizosaccharomyces pombe* cells were grown at 32°C to exponential phase ($OD_{600} = 0.5$) according to standard procedures [49] in either complex yeast extract medium with

supplements (YES) or Edinburgh minimal medium (EMM). Routine optical density (OD) measurements of liquid cell cultures were taken using the WPA CO 8000 Cell Density Meter (Biochrom). Growth rate measurements and calculation of cell culture doubling time (DT) were performed as described previously [20].

The construction of the scarless knock-in strain expressing C-terminally TAP-tagged Cbf11 from its endogenous chromosomal locus (Cbf11-TAP) was described previously [19].

In this study, we N-terminally TAP-tagged *mga2* at its chromosomal locus to allow detection and distinction between the full-length precursor form and the cleaved active N-terminal transcription factor form of Mga2. To this end, the CRISPR/Cas9-based *SpEDIT* strategy was adapted from [48]. WT cells (JB32) were synchronized in G1 and transformed [50] with the Cas9/sgRNA_*mga2* plasmid (pMP167) together with a fragment of the *TAP-mga2** sequence (plasmid pMP168 digested by *SacI* and *XhoI*) as template for homologous recombination. The *TAP-mga2** sequence contained a silent mutation which prevented re-cleavage of the tagged *mga2* locus by Cas9. After selection on YES plates containing 100 μg/ml of Nourseothricin (NAT), the smallest NAT-resistant colonies were re-streaked onto non-selective YES plates to allow loss of the Cas9/sgRNA_*mga2* plasmid. Correct integration of the *TAP-mga2** sequence was verified by PCR (primers MaP172 and AJ92) and sequencing. Expression of the TAP-Mga2 protein was verified by western blot with an anti-TAP antibody (Thermo Scientific, CAB1001).

Constructions of the *Δcbf11* and *Δmga2* single deletion strains were published elsewhere [16,19]. To create the *Δmga2 Δcbf11*, Cbf11-TAP *Δmga2*, and TAP-Mga2 *Δcbf11* strains, the pClone system was used for deletion of the *cbf11* and/or *mga2* genes [51].

The lists of strains, plasmids and primers used in this study are provided in Tables A-C in S1 Data.

## Spot tests

Exponentially growing cell cultures were 10-fold serially diluted and spotted onto control YES plates and testing YES plates containing thiabendazole (TBZ; 15 and 18 μg/ml) or camptothecin (CPT; 4 and 6 μM) for drug sensitivity assays. Plates were imaged after 6 days of incubation at 32˚C. Two independent experiments with technical replicates were performed.

## Colony-forming unit assay

The CPT and TBZ sensitivity of the WT, *Δcbf11*, *Δmga2,* and *Δmga2 Δcbf11* cells was quantitatively examined by counting colony-forming units (CFU) upon exposure to the stressors. A total of 100–200 μl of exponentially growing cultures, diluted between 7500 and 37500x to give rise to typically 200–600 CFU per plate, were plated in triplicate onto control YES plates and testing YES plates containing 6 μM CPT or 13 μg/ml TBZ using glass 4.5 mm beads. Plates were incubated for 7–17 days at 32˚C and formed colonies were counted manually. 3–4 independent experiments were performed. To determine statistical significance, Welch Two Sample t-test was performed on normalized data (t.test function in R, paired = FALSE, alternative = "less").

## Fluorescence microscopy

Fluorescence microscopy was performed as described previously [21]. Briefly, for observation of nuclear morphology and quantification of 'cut' phenotype, nuclei of ethanol-fixed cells were stained with DAPI. 'Cut' cells were counted manually; 200–1000 cells per sample were analyzed. For visualization of lipid droplets (LDs), live cells grown to exponential phase in YES or EMM were stained with BODIPY 493/503. 7–14 images per sample were processed by an

automated MATLAB pipeline [52]. Resulting output data were processed using R. At least six ('cut' counting) or three (LD content) independent experiments were performed. To determine statistical significance, Welch Two Sample t-test was performed on raw data (t.test function in R, paired = FALSE, alternative = "greater" for 'cut' counting, alternative = "less" for LD content).

### RT-qPCR

Expression of selected lipid metabolism genes in exponentially growing cells was analyzed as described previously [21]. Briefly, total RNA was extracted and converted to cDNA using random priming. qPCR reactions were performed in technical triplicates. For normalization, *act1* (actin) and *rho1* (Rho1 GTPase) were used as internal reference genes. One-sided Mann-Whitney U test was used to determine statistical significance (wilcox.test function in R, paired = FALSE, alternative = "less"). At least three independent biological experiments were performed. The primers used are listed in Table C in S1 Data.

### RNA-seq

RNA-seq analyses for WT (JB32) and *Δcbf11* (MP44) were already published [21]. Additional RNA-seq analyses shown in this study were performed as described previously [21]. Briefly, *Δmga2* (MP815), *Δmga2 Δcbf11* (MP836), and *Pcut6MUT* (MP636) cells were cultured in YES to exponential phase and harvested. For cerulenin-related analyses, WT cells were grown to exponential phase and treated with DMSO (control) or 20 μM cerulenin for 1 hour in YES and then harvested. Samples were prepared from 3 biological replicates. Total RNA was isolated, treated with TURBO DNase and column-purified. RNA quality was assessed on Agilent Bioanalyzer. Sequencing library construction and sequencing were performed by the Genomics and Bioinformatics service facility at the Institute of Molecular Genetics (Czech Academy of Sciences, Prague) using the Illumina NextSeq 500 sequencing system. For detailed RNA-seq data analysis please refer to [21].

### ChIP-nexus

ChIP-nexus analyses for Cbf11-TAP (MP707) and untagged WT (JB32, a negative control) were already published elsewhere [21]. Three independent biological experiments of ChIP-nexus for Cbf11-TAP *Δmga2* (MP921), TAP-Mga2 (MP872), and TAP-Mga2 *Δcbf11* (MP915) presented here were performed and analyzed in the same manner as described previously for Cbf11-related analyses [21]. Briefly, cells were cultivated in YES to exponential phase and fixed with 1% formaldehyde followed by quenching in 125 mM glycine. Chromatin was extracted and sheared with the Bioruptor sonicator (Diagenode). For the intended downstream normalization of ChIP samples, 2 mg of each *S. pombe* chromatin were mixed with 40,000-fold diluted *Saccharomyces cerevisiae* chromatin sample containing TAP-tagged version of histone H2A.2 (strain MP792). Note that in the end the spike-in normalization was disregarded during data analyses. Chromatin immunoprecipitation was performed overnight at 4°C with 100 μl of BSA-blocked Pan Mouse IgG Dynabeads (Invitrogen, 110.41). The precipitated material was ChIP-exo treated, decrosslinked, and treated with RNase A and proteinase K. DNA was purified using phenol-chloroform extraction followed by sodium acetate/ethanol precipitation and was used for the subsequent construction of ChIP-nexus sequencing libraries. Sequencing was performed by the Genomics and Bioinformatics service facility at the Institute of Molecular Genetics (Czech Academy of Sciences, Prague) using the Illumina NextSeq 500 sequencing system. The list of used ChIP-nexus oligonucleotides is provided in Table D in S1 Data. For detailed ChIP-nexus data analysis please refer to [21].

## Proteomic analysis

WT (JB32), *Δcbf11* (MP44), and *Δmga2* (MP815) cells were cultured in YES to exponential phase in three independent biological replicates. 20 ml of culture were harvested and the cell pellets were flash-frozen in liquid nitrogen. Samples were processed in the Laboratory of Mass Spectrometry at Biocev Research Center (Faculty of Science, Charles University, Prague), where the bottom-up untargeted proteomic analysis was performed.

To extract proteins, cells were lysed by milling using FastPrep (MP Biomedicals) with glass beads and incubated at 95˚C for 10 min in 100 mM Triethylammonium bicarbonate (TEAB) containing 2% sodium deoxycholate (SDC), 40 mM chloroacetamide, and 10 mM Tris(2-carboxyethyl)phosphine (TCEP). Protein concentration was determined using BCA protein assay kit (Thermo Scientific) and 30 μg of protein per sample were further processed using the SP3 technology for protein cleanup and digestion [53]. Briefly, 5 μl of SP3 beads were added to protein samples and protein binding was induced by adding ethanol to 60% (vol./vol.) final concentration. Samples were mixed and incubated for 5 min at room temperature. Using a magnetic rack, the unbound material was discarded and the beads were subsequently washed with 80% ethanol. Next, samples were digested with trypsin (1:30 enzyme to protein ratio) and reconstituted in 100 mM TEAB at 37˚C overnight. The resulting peptide samples were acidified with trifluoroacetic acid (1% final concentration), desalted using in-house stage tips packed with C18 disks (Empore) according to [54], and subjected to nano-liquid chromatography coupled to tandem mass spectrometry (nLC-MS2) analysis [55]. Nano Reversed phase columns (EASY-Spray column, 50 cm x 75 μm ID, PepMap C18, 2 μm particles, 100 Å pore size) were used. Mobile phase buffer A was composed of water and 0.1% formic acid. Mobile phase B was composed of acetonitrile and 0.1% formic acid. Samples were loaded onto the trap column (C18 PepMap100, 5 μm particle size, 300 μm x 5 mm, Thermo Scientific) for 4 min at 18 μl/min. Loading buffer was composed of water, 2% acetonitrile and 0.1% trifluoroacetic acid. Peptides were eluted with Mobile phase B gradient from 4% to 35% B in 60 min. Eluting peptide cations were converted to gas-phase ions by electrospray ionization and analyzed on a Thermo Orbitrap Fusion (Q-OT-qIT, Thermo Scientific). Survey scans of peptide precursors from 350 to 1400 *m/z* were performed in orbitrap at 120K resolution (at 200 *m/z*) with a $5\times10^5$ ion count target. Tandem MS was performed by isolation at 1.5 Th with the quadrupole, HCD fragmentation with normalized collision energy of 30, and rapid scan MS analysis in the ion trap. The MS2 ion count target was set to $10^4$ and the max injection time was 35 ms. Only those precursors with charge state 2–6 were sampled for MS2. The dynamic exclusion duration was set to 45 s with a 10 ppm tolerance around the selected precursor and its isotopes. Monoisotopic precursor selection was turned on. The instrument was run in top speed mode with 2 s cycles.

All data were analyzed and quantified with the MaxQuant software (version 2.0.3.0) [56]. The false discovery rate (FDR) was set to 1% for both proteins and peptides. A minimum peptide length was specified to seven amino acids. The Andromeda search engine was used for the MS/MS spectra search against the *S. pombe* database (downloaded from uniprot.org). Enzyme specificity was set as C-terminal to Arg and Lys, also allowing cleavage at proline bonds and a maximum of two missed cleavages. Carbamidomethylation of cysteine was selected as a fixed modification and N-terminal protein acetylation and methionine oxidation as variable modifications. The "match between runs" feature of MaxQuant was used to transfer identifications to other LC-MS/MS runs based on their masses and retention time (maximum deviation 0.7 min) and this was also used in quantification experiments. Quantifications were performed with the label-free algorithm in MaxQuant [57]. Data analysis was performed using Perseus 1.6.15.0 software [58]. Proteins showing at least 2-fold and statistically significant change in expression in *Δcbf11* or *Δmga2* mutants compared to WT were selected for further analyses.

### Phylogenetic distribution of Cbf11 and Mga2 homologs

The Protein BLAST (*blastp*) tool (https://blast.ncbi.nlm.nih.gov/Blast.cgi?PROGRAM=blastp&PAGE_TYPE=BlastSearch&LINK_LOC=blasthome) from the National Centre for Biotechnology Information (NCBI) was used to determine the distribution of Cbf11 and Mga2 homologs among Fungi (taxID: 4751). The *blastp* search was performed using default settings against the Reference sequence database (Refseq_protein; https://www.ncbi.nlm.nih.gov/refseq/; June 22, 2023) with the protein sequences of *S. pombe* Cbf11 (SPCC736.08) and Mga2 (SPAC26H5.05) taken from the PomBase database (https://www.pombase.org/) as queries. In total, 226 sequences were found for the Cbf11 query and 1223 sequences were found for the Mga2 query. These protein sequences were manually filtered to only keep those that showed high-quality alignment in the regions of the known signature domains of Cbf11 (RHR-N and Beta-trefoil) and Mga2 (IPT and Ankyrin repeats). As a result, three lists of taxIDs were compiled: taxons containing both Cbf11 and Mga2 homologs, taxons containing only Cbf11 homolog(s), and taxons containing only Mga2 homolog(s). These lists were visualized in the NCBI Common Taxonomy Tree tool (https://www.ncbi.nlm.nih.gov/Taxonomy/CommonTree/wwwcmt.cgi). Subsequently, the taxons in which homologs from only one of the Cbf11/Mga2 families were originally found were searched by *blastp* individually to confirm the absence of the respective protein family (performed July 2–20, 2023). Thereby, three final taxID lists were created (Cbf11&Mga2, Cbf11_only, Mga2_only). A combined list of all identified taxIDs was used to generate a phylogenetic tree by the NCBI Common Taxonomy Tree and individual taxIDs were assigned the corresponding taxon names using the NCBI TaxIdentifier tool (https://www.ncbi.nlm.nih.gov/Taxonomy/TaxIdentifier/tax_identifier.cgi). iTOL v6 was used to display and annotate the final phylogenetic tree (https://itol.embl.de/). The taxIDs lists with corresponding taxon names are provided in Table E in S1 Data.

## Supporting information

**S1 Fig. The *Δcbf11* and *Δmga2* mutants show similar, non-additive growth defects.** These defects are partially suppressed in the EMM minimal medium. Mean ± SD values, as well as individual data points for ≥2 independent experiments are shown.
(TIFF)

**S2 Fig. RNA-seq shows non-additive decrease in transcript level of lipid metabolism genes in *Δmga2 Δcbf11* cells.** The *ptl1* and *ptl2* genes encode triacylglycerol lipases, the *fas1* gene encodes the fatty acid synthase alpha subunit. Mean values for 3 independent experiments are shown.
(TIFF)

**S3 Fig. Comparisons of sets of ncRNAs showing differential expression upon various perturbations of lipid metabolism.** Sets of ncRNAs upregulated (top panel) or downregulated (bottom panel) in the *Δcbf11*, *Δmga2*, *Δmga2 Δcbf11* or *Pcut6MUT* mutants, or upon cerulenin treatment were subdivided based on their mutual overlaps. The sizes of the intersections were plotted, with subset membership indicated below the plot. The results show that hundreds of ncRNAs are jointly upregulated under all conditions tested. The figure was created using the UpSetR package in R [59].
(TIFF)

**S4 Fig. Cells lacking Cbf11 and/or Mga2 show similar sensitivity to camptothecin (CPT) and thiabendazole (TBZ).** Quantification of colony-forming units upon chronic exposure to indicated concentrations of CPT and TBZ is shown as % survival. Mean ± SD values, as well as

individual data points for 3–4 independent experiments are shown. Significance was determined by an unpaired one-sided Welch Two Sample t-test.
(TIFF)

**S5 Fig. The doubling times of *cbf11* and *mga2* knock-outs and scarless knock-ins.**
Mean ± SD values, as well as individual data points for ≥2 independent experiments are
shown.
(TIFF)

**S6 Fig. ChIP-nexus analysis of Cbf11 and Mga2 binding to promoters of lipid metabolism
genes. (A)** Promoters to which both Cbf11 and Mga2 bind, and the binding is Cbf11-dependent. **(B)** Promoters where binding shows varying dependence on Cbf11 and/or Mga2. **(C)**
Control loci with no binding. Mean strand-specific coverage profile of 3 independent experiments for Cbf11 and Mga2 in the indicated genetic backgrounds, and a strand-specific coverage profile for untagged WT cells (negative control) are visualized in the Integrated Genome
Viewer (IGV; Broad Institute). Gene orientation is indicated by three arrowheads; the maximum Y-axis value for each track is indicated on the right side of each panel.
(TIFF)

**S1 Text. *TAP-mga2*\* template for HR.**
(DOCX)

**S1 Data. Table A—List of strains, Table B—List of plasmids, Table C—List of oligos,
Table D—ChIP-nexus oligos, Table E—Phylogen. distrib.**
(XLSX)

**S2 Data. Source numerical data.**
(XLSX)

## Acknowledgments

We would like to thank Barbora Huraiová and Silvia Bágeľová-Poláková (Slovak Academy of
Science, Bratislava) for providing plasmid DNA, and Marian Novotný for help with interpreting protein 3D structure predictions. We are grateful to Štěpánka Hrdá and Blanka Hamplová
from the Genomics core facility at Biocev Research Center, Faculty of Science, Charles University for Sanger sequencing and quality control of nucleic acids; Adéla Kracíková, Kateřina Svobodová and Eva Krellerová for excellent technical assistance; all members of the GenoMik
group for their support and insightful discussions. Microscopy was performed in the Vinicna
Microscopy Core Facility co-financed by the Czech-BioImaging large RI project LM2023050.
Computational resources were supplied by the project "e-Infrastruktura CZ" (e-INFRA
LM2018140) provided within the program Projects of Large Research, Development and Innovations Infrastructures. LC-MS analyses were performed in the Laboratory of Mass Spectrometry at Biocev Research Center; Faculty of Science, Charles University.

## Author Contributions

**Conceptualization:** Anna Marešová, Martin Převorovský.

**Data curation:** Anna Marešová.

**Formal analysis:** Anna Marešová, Michaela Grulyová, Viacheslav Zemlianski, Jarmila Princová, Martin Převorovský.

**Funding acquisition:** Anna Marešová.

**Investigation:** Anna Marešová, Michaela Grulyová, Viacheslav Zemlianski, Jarmila Princová.

**Methodology:** Anna Marešová, Miluše Hradilová.

**Project administration:** Anna Marešová, Martin Převorovský.

**Resources:** Miluše Hradilová.

**Supervision:** Martin Převorovský.

**Visualization:** Anna Marešová, Martin Převorovský.

**Writing – original draft:** Anna Marešová, Martin Převorovský.

**Writing – review & editing:** Anna Marešová, Martin Převorovský.

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
