## [Decision Letter · Decision Letter 0]

20 Nov 2024

Dear Dr Převorovský,

We are pleased to inform you that your manuscript entitled "Cbf11 and Mga2 function together to activate transcription of lipid metabolism genes and promote mitotic fidelity in fission yeast" has been editorially accepted for publication in PLOS Genetics. Congratulations!

Yours sincerely,

Cathy Savage-Dunn

Academic Editor

PLOS Genetics

Monica Colaiácovo

Section Editor

PLOS Genetics

Aimée Dudley

Editor-in-Chief

PLOS Genetics

Anne Goriely

Editor-in-Chief

PLOS Genetics

Comments from the reviewers (if applicable):

Reviewer's Responses to Questions

**Comments to the Authors:**

Reviewer #1: In this study, Maresova and colleagues showed several lines of evidence indicating that Cbf11 and Mga2 in S. pombe function in the same pathway regulating transcription of genes for lipid metabolism and mitotic fidelity. Based on their observations, the authors proposed a mechanistic model in which Cbf11 and Mga2 may function cooperatively, either as a single protein complex or without requiring physical interaction. They also examined the evolutionary distribution of Cbf11 and Mga2 homologs among fungi and suggested that transcriptional regulation of lipid metabolism genes has been remodeled during evolution.

In this revised manuscript, additional experiments have been appropriately performed and the text has been satisfactorily revised. The reviewers believe that the revised manuscript is suitable for publication.

Reviewer #2: The revision has fully addressed my concerns. I support the publication of this paper.

**Have all data underlying the figures and results presented in the manuscript been provided?**

Reviewer #1: Yes

Reviewer #2: Yes

PLOS authors have the option to publish the peer review history of their article (what does this mean?). If published, this will include your full peer review and any attached files.

Reviewer #1: No

Reviewer #2: No

**Data Deposition**

http://datadryad.org/submit?journalID=pgenetics&manu=PGENETICS-D-24-01245

**Press Queries**

---

## [Editor Report · Acceptance letter]

2 Dec 2024

PGENETICS-D-24-01245 

Cbf11 and Mga2 function together to activate transcription of lipid metabolism genes and promote mitotic fidelity in fission yeast 

Dear Dr Převorovský, 

We are pleased to inform you that your manuscript entitled "Cbf11 and Mga2 function together to activate transcription of lipid metabolism genes and promote mitotic fidelity in fission yeast" has been formally accepted for publication in PLOS Genetics! Your manuscript is now with our production department and you will be notified of the publication date in due course.

With kind regards,

Anita Estes

PLOS Genetics

On behalf of:
